


# Measurement report: Size-resolved mass concentration of equivalent black carbon-containing particle larger than 700 nm and its role in radiation

Weilun Zhao[1], Ying Li[2,3], Gang Zhao[4], Song Guo[4], Nan Ma[5], Shuya Hu[4], Chunsheng Zhao[1]

[1]Department of Atmospheric and Oceanic Sciences, School of Physics, Peking University, Beijing 100871, China

[2]Department of Ocean Science and Engineering, Southern University of Science and Technology, Shenzhen 518055, China

[3]Southern Marine Science and Engineering Guangdong Laboratory, Guangzhou 511458, China

[4]State Key Joint Laboratory of Environmental Simulation and Pollution Control, College of Environmental Sciences and Engineering, Peking University, Beijing 100871, China

[5]Institute for Environmental and Climate Research, Jinan University, Guangzhou 511443, China

*Correspondence to*: Chunsheng Zhao (zcs@pku.edu.cn)

**Abstract.** Black carbon (BC) mass size distribution (BCMSD) is crucial in both environment and climate system due to BC's intense size-dependent absorption of solar radiation. BC-containing particles of size larger than 700 nm ($BC_{>700}$) could contribute to larger than half of bulk BC mass concentration. Unfortunately, previous methods concentrated on BC-containing particles less than 700 nm because of technical limitation. The contribution of BC to absorption and radiative effect would be underestimated without consideration of $BC_{>700}$. In this study, equivalent BCMSD (eBCMSD) from 150 nm up to 1.5 μm was measured at high time resolution of 1 h for the first time by an aerodynamic aerosol classifier in tandem with an aethalometer in two field campaigns over eastern China, namely Changzhou located in the Yangtze River Delta and Beijing located in the North China Plain. The results revealed that the level of eBCMSD in both Changzhou and Beijing increased with increasing pollution. The pattern of eBCMSD in Changzhou (Beijing) was mostly bimodal (unimodal) peaking at 240 and 1249 nm (427 nm). The peak diameter of eBCMSD in Changzhou did not shift significantly with increasing pollution (240 to 289 nm). In contrast, the peak diameter of eBCMSD in Beijing shifted towards larger size from 347 to 527 nm with increasing pollution, indicating the aging process in urban site was different from that in regional background site. eBCMSD in both Changzhou and Beijing had significant diurnal cycle with lower (higher) level of eBCMSD during daytime (nighttime). Equivalent $BC_{>700}$ ($eBC_{>700}$) was ubiquitous and varied significantly with different locations and pollution levels. The campaign-averaged contribution of $eBC_{>700}$ to bulk eBC mass concentration ($m_{eBC,bulk}$), bulk absorption coefficient ($\sigma_{ab,bulk}$) as well as estimated direct radiative forcing of eBC ($DRF_{eBC}$) in Changzhou and Beijing were 27.8 (20.9 ~ 36.5) % and 24.1 (17.5 ~ 34.2) %, 19.6 (15.8 ~ 24.6) % and 25.9 (19.6 ~ 33.7) %, as well as 20.5 (18.4 ~ 22.2) % and 21.0 (16.3 ~ 26.1) %, respectively. $m_{eBC,bulk}$, $\sigma_{ab,bulk}$ as well as $DRF_{eBC}$ of $eBC_{>700}$ in Changzhou (Beijing) varied by 3.6 (5.1) times from 0.11 (0.07) to 0.40 (0.36) μg m$^{-3}$, 3.2 (5.5) times from 0.54 (0.63) to 1.75 (3.45) Mm$^{-1}$ as well as 2.4 (4.7) times from 0.1 (0.1) to 0.24 (0.47) W m$^{-2}$, respectively, with the aggravation of pollution. The contribution of $eBC_{>700}$ to $m_{eBC,bulk}$ and $\sigma_{ab,bulk}$ had significant diurnal cycle with higher



(lower) fraction during daytime (nighttime) in both Changzhou and Beijing. A case study indicated that the contribution of
$eBC_{>700}$ to $m_{eBC,bulk}$, $\sigma_{ab,bulk}$ and $DRF_{eBC}$ could reach up to 50 %, 50 % and 40 %, respectively. It was highly recommended to
consider whole size range of BC-containing particles in the model estimation of BC radiative effect.
**1 Introduction**

Black carbon (BC) is strong light-absorbing carbonaceous particle (Bond and Bergstrom, 2006) from incomplete

combustion of fossil fuel or biomass (Bond et al., 2004). Absorption of BC increases light extinction (Moosmuller et al., 2009)
and has warming effect on the climate system (Bond, 2001). BC radiative effect had considerable uncertainties and different
estimated BC radiative effects did not even converge to same order of magnitude (Bond et al., 2013;Szopa et al., 2021).

Previous estimation of BC radiative effect was based on bulk BC mass concentration ($m_{BC,bulk}$) from emission inventory

and prescribed mass absorption cross section (MAC) (Bond et al., 2013). Both $m_{BC,bulk}$ and MAC was influenced by BC mass
size distribution (BCMSD). BCMSD was one of the BC microphysical properties that BC radiative effect was highly sensitive
to (Matsui et al., 2018), and could result in obvious variation in aerosol radiative forcing (Zhao et al., 2019). BCMSD
depended on the emission source essentially. For example, the peak diameter of freshly emitted BCMSD from fossil fuel was
generally smaller than that from biomass burning (Berner et al., 1984;Artaxo et al., 1998;Schwarz et al., 2008). After BC was
emitted to the ambient environment, BCMSD was influenced by BC aging process, during which BC optical properties
underwent remarkable changes (Zhang et al., 2008). For instance, BC could be coated by other non-BC materials during
atmospheric transport. The existence of non-BC coating enhanced BC absorption and the phenomenon was termed as "lensing
effect" (Fuller et al., 1999), of which the accurate quantification was a critical challenge in estimating BC radiative effect (Liu
et al., 2017). The information of BCMSD was required to resolve the influence of "lensing effect" on BC radiative forcing.

Guo (2016) reported that reported that elemental carbon (EC, Petzold et al. (2013)) containing particles larger than 2.1 μm

accounted for 27.6 ~ 35.2 % of bulk EC mass concentration ($m_{EC,bulk}$). Wang et al. (2017) reported that EC-containing particle
larger than 1.1 μm accounted for 40.6 ~ 65.5 % of $m_{EC,bulk}$. Wang et al. (2022) indicated that EC-containing particle larger
than 1 μm contributed to 50 ~ 54 % of $m_{EC,bulk}$. Therefore, BC-containing particle larger than 1 μm contributed to significant
part of total BC mass. Wang et al. (2022) found that these super large carbon-containing particles were super-aggerated BC
particles with fractal structure or BC-containing particles with massive coating from secondary processes. It should be noted
that current characterization of BC-containing particle larger than 1 μm could be only achieved through EC mass size
distribution (ECMSD) measurement by off-line thermo/optical organic carbon/elemental carbon analysis of size-segregated
filter-based samples (Chow et al., 2001). The resulting time-resolution of ECMSD was 24 ~ 48 h. Considering that the typical
time scale of BC aging was 4 ~ 18 h (Peng et al., 2016), current measured ECMSD could not resolve atmospheric aging of
BC-containing particles larger than 1 μm. Actually, current method capable of measuring BC-containing particle on time scale
of BC aging, namely laser-induced incandescence technique (Schwarz et al., 2006), was limited to size less than 700 nm. The
characterization of BC-containing particles larger than 700 nm ($BC_{>700}$) during atmospheric aging was still unclear. The





contribution of BC$_{>700}$ to absorption and BC radiative forcing was lack of study.
In this study, equivalent BC (eBC, Petzold et al. (2013)) mass size distribution (eBCMSD) up to 1.5 μm was measured with
a time resolution of 1 h to study the evolution of equivalent BC$_{>700}$ (eBC$_{>700}$) as well as the contribution of eBC$_{>700}$ to bulk
eBC mass concentration ($m_{eBC,bulk}$), bulk absorption coefficient ($\sigma_{ab,bulk}$) and eBC direct radiative forcing. eBCMSD was
determined by an aerosol aerodynamic classifier (AAC, Cambustion, UK, Tavakoli and Olfert (2013)) in tandem with an
aethalometer (model AE33, Magee, USA, Drinovec et al. (2015), AAC – AE33) based on the method proposed by Zhao et al.
(2022). eBCMSD was measured in two different locations of eastern China to study the spatial difference of eBC$_{>700}$. Direct
radiative forcing of eBC (DRF$_{eBC}$) was estimated by the Santa Barbara DISORT (discrete ordinates radiative transfer)
Atmospheric Radiative Transfer (SBDART) model (Ricchiazzi et al., 1998).
The structure of this study was organized as follows. Section 2 introduced the field measurement, instrumental setup, and
details about estimation of DRF$_{eBC}$. Section 3 discussed the evolution as well as mass, absorption and radiation contribution
of eBC$_{>700}$ based on the field measurement. Section 4 came to the conclusions.
**2 Methods**
**2.1 Field measurement**
The AAC-AE33 system was first applied to a field measurement in Changzhou, Jiangsu Province, China (119°36′E, 31°43′
N), situated at the Yangtze River Delta, from May 17$^{th}$ to June 3$^{rd}$ in 2021. Then, the AAC-AE33 was deployed in Beijing,
China (116°18′E, 39°59′N), located in the North China Plain, from October 29$^{th}$ 2021 to January 25$^{th}$ 2022. The measurement
station in Changzhou was a typical regional background site and the other in Beijing was representative of urban environment.
The detailed description of Changzhou and Beijing could be found in Zhao et al. (2022) and Zhao et al. (2019), respectively.
**2.2 Instrumental setup**
The instrumental setup for eBCMSD measurement was illustrated in detail by Zhao et al. (2022) and introduced here briefly.
As shown in Fig. 1, a PM$_{10}$ inlet (16.67 L min$^{-1}$) was used to sample ambient aerosol particles. Then particles passed through
a silica gel diffusion drier, where relative humidity (RH) was decreased to less than 30 %, before sampled by the AAC-AE33.
AAC-AE33 measured size-resolved absorption coefficient ($\sigma_{ab,size-resolved}$) at a flow rate of 3 L min$^{-1}$ in Changzhou and 2 L
min$^{-1}$ in Beijing, respectively. AAC was set to scan 12 logarithmically equally distributed aerodynamic sizes ranging from
200 nm to 1.5 μm in Changzhou and 150 nm to 1.5 μm in Beijing, respectively. It should be pointed out that particle diameter
($D_p$) was aerodynamic size in this study. Particles of each scanned size were sampled for 5 min, so the time resolution of
$\sigma_{ab,size-resolved}$ came to 1 h. The measured $\sigma_{ab,size-resolved}$ at wavelength of 880 nm by AE33 was used to derive eBCMSD because
BC was the major contributor of aerosol absorption at 880 nm (Ramachandran and Rajesh, 2007).
MAC was required to convert absorption coefficient to eBC mass concentration. The size-dependent MAC was modeled
based on the scheme proposed by Zhao et al. (2021), which required size-resolved particle number concentration ($N_{size-resolved}$).
$N_{size-resolved}$ was measured by a scanning mobility particle sizer (SMPS, TSI, USA) at 0.3 L min$^{-1}$ as well as an aerodynamic





particle sizer (APS, TSI, USA) at 5 L min$^{-1}$ in Changzhou and an AAC in tandem with condensation particle counter (CPC,
TSI, USA, AAC – CPC, Johnson et al. (2018)) at 1 L min$^{-1}$ in Beijing, respectively. AAC-AE33 measured $\sigma_{ab,size\text{-}resolved}$ and
determined eBCMSD synchronously. Therefore, the contribution of eBC$_{>700}$ to both bulk absorption and $m_{eBC,bulk}$ could be
quantified simultaneously.
In this study, the bulk mass concentration of eBC-containing particle ($m_{eBC,bulk}$) was defined as
$$m_{eBC,bulk} = \int_{200\,nm}^{1500\,nm} \frac{dm_{eBC}}{dlogD_p} dlogD_p, \tag{1}$$
where $\frac{dm_{eBC}}{dlogD_p}$ was eBCMSD, and the lower limit of integral was 200 nm in both Changzhou and Beijing for the convenience
of comparison. The difference of 50 nm in $D_{p0}$ had little influence the conclusion of this study. The bulk mass concentration
of eBC$_{>700}$ ($m_{eBC,bulk,>700}$) was defined as
$$m_{eBC,bulk,>700} = \int_{700\,nm}^{1500\,nm} \frac{dm_{eBC}}{dlogD_p} dlogD_p. \tag{2}$$
The contribution of eBC$_{>700}$ to $m_{eBC,bulk}$ ($f_{m,>700}$) was defined as
$$f_{m,>700} = \frac{m_{eBC,bulk,>700}}{m_{eBC,bulk}} \times 100\ \%. \tag{3}$$
Similarly, the bulk absorption coefficient ($\sigma_{ab,bulk}$) was defined as
$$\sigma_{ab,bulk} = \int_{200\,nm}^{1500\,nm} \frac{d\sigma_{ab}}{dlogD_p} dlogD_p, \tag{4}$$
where $\frac{d\sigma_{ab}}{dlogD_p}$ was $\sigma_{ab,size\text{-}resolved}$. The bulk absorption coefficient of eBC$_{>700}$ ($\sigma_{ab,bulk,>700}$) was defined as
$$\sigma_{ab,bulk,>700} = \int_{700\,nm}^{1500\,nm} \frac{d\sigma_{ab}}{dlogD_p} dlogD_p. \tag{5}$$
The contribution of eBC$_{>700}$ to $\sigma_{ab,bulk}$ ($f_{ab,>700}$) was defined as
$$f_{ab,>700} = \frac{\sigma_{ab,bulk,>700}}{\sigma_{ab,bulk}} \times 100\ \%. \tag{6}$$
**2.3 Estimation of direct radiative forcing of equivalent black carbon**
The direct radiative effect was one of the BC characteristics that arouse extensive concerns. The SBDART model was
employed to study the characteristics of DRF$_{eBC}$. Specifically, the instantaneous DRF$_{eBC}$ was estimated at the top of
atmosphere (TOA) under the cloud-free condition. Wavelengths from 250 nm to 4 μm were simulated in this study. Direct
radiative forcing of aerosol (DRF$_{aerosol}$) was defined as (Zhao et al., 2018):
$$DRF_{aerosol} = \left(F_{aerosol,\downarrow} - F_{aerosol,\uparrow}\right) - \left(F_{clearsky,\downarrow} - F_{clearsky,\uparrow}\right), \tag{7}$$
where $F_{aerosol,\downarrow}$ ($F_{aerosol,\uparrow}$) was downward (upward) radiative irradiance flux at TOA with aerosol, and $F_{clearsky,\downarrow}$
($F_{clearsky,\uparrow}$) was downward (upward) radiative irradiance flux at TOA without aerosol. Direct radiative forcing of aerosol
without eBC (DRF$_{aerosol,noneBC}$) was defined as:
$$DRF_{aerosol,noneBC} = \left(F_{aerosol,noneBC,\downarrow} - F_{aerosol,noneBC,\uparrow}\right) - \left(F_{clearsky,\downarrow} - F_{clearsky,\uparrow}\right), \tag{8}$$
where $F_{aerosol,noneBC,\downarrow}$ ($F_{aerosol,noneBC,\uparrow}$) was downward (upward) radiative irradiance flux at TOA with aerosol except eBC.



The $\text{DRF}_{\text{eBC}}$ was defined as the difference between $\text{DRF}_{\text{aerosol}}$ and $\text{DRF}_{\text{aerosol,noneBC}}$:
$$\text{DRF}_{\text{eBC}} = \left(F_{\text{aerosol},\downarrow} - F_{\text{aerosol},\uparrow}\right) - \left(F_{\text{aerosol,noneBC},\downarrow} - F_{\text{aerosol,noneBC},\uparrow}\right). \tag{9}$$
Similarly, the direct radiative forcing of $\text{eBC}_{>700}$ ($\text{DRF}_{\text{eBC},>700}$) was defined as:
$$\text{DRF}_{\text{eBC},>700} = \left(F_{\text{aerosol},\downarrow} - F_{\text{aerosol},\uparrow}\right) - \left(F_{\text{aerosol,noneBC},>700,\downarrow} - F_{\text{aerosol,noneBC},>700,\uparrow}\right), \tag{10}$$
where $F_{\text{aerosol,noneBC},>700,\downarrow}$ ($F_{\text{aerosol,noneBC},>700,\uparrow}$) was downward (upward) radiative irradiance flux at TOA with aerosol
except $\text{eBC}_{>700}$. The contribution of $\text{eBC}_{>700}$ to $\text{DRF}_{\text{eBC}}$ ($f_{\text{DRF},>700}$) was defined as
$$f_{\text{DRF},>700} = \frac{\text{DRF}_{\text{eBC},>700}}{\text{DRF}_{\text{eBC}}} \times 100 \ \%. \tag{11}$$
SBDART simulation required information of surface albedo, vertical profiles of meteorological parameters and aerosol
optical parameters. Surface albedo was acquired from Moderate Resolution Imaging Spectroradiometer (MODIS)/Terra
surface reflectance data with temporal and spatial resolution of 1 d and 0.05° (MOD09CMG). The gridded data around the
measurement site was averaged to represent surface albedo of the measurement site.
The vertical profile of meteorological parameters included vertical profile of pressure, temperature, water vapor and ozone,
which were obtained from the fifth generation ECMWF (European Center for Medium Range Weather Forecasts) reanalysis
data for global climate and weather (ERA5). The ERA5 data had temporal and spatial resolution of 1 h and 0.25° with 38
vertical layers from surface to about 50 km above surface. At each layer, the gridded data around the measurement site was
also averaged to represent meteorological parameters of the measurement site. The time resolution of meteorological
parameters was averaged to daily to match that of surface albedo.
The vertical profile of aerosol optical parameters included the vertical profile of bulk aerosol extinction coefficient ($\sigma_{\text{ext,bulk}}$),
single scattering albedo (SSA) and asymmetry factor (g) at different wavelengths, which were parameterized based on the
study of Zhao et al. (2019) and described here briefly. The bulk aerosol particle number concentration ($N_{\text{bulk}}$) was
parameterized according to aircraft study by Liu et al. (2009). Dry $N_{\text{size-resolved}}$ at different heights had the same shape after
normalized by corresponding $N_{\text{bulk}}$. The parameterization of $m_{\text{eBC,bulk}}$ and eBCMSD was the same as $N_{\text{bulk}}$ and dry $N_{\text{size-resolved}}$.
51% of eBC mass was assumed externally mixed and the rest of eBC mass was assumed internally mixed with core-shell
geometry (Ma et al., 2012) in each size bin. For the case of aerosol without eBC-containing particle (larger than 700 nm),
eBCMSD (larger than 700 nm) was set to 0. The aerosol optical parameters varying with height-dependent RH were calculated
by Mie scattering theory and κ-Kohler theory (Petters and Kreidenweis, 2007) assuming hygroscopic growth parameter of
0.22 (Tan et al., 2019). The refractive indices of eBC, water and non-eBC material without water were assumed $1.8 + 0.54i$
(Kuang et al., 2015), $1.33 + 10^{-7}i$ and $1.53 + 10^{-7}i$ (Wex et al., 2002), respectively. The refractive index of non-eBC material
mixed with water after hygroscopic growth was derived by volume-weighted rule (Wex et al., 2002). With the above
information, the vertical profiles of $\sigma_{\text{ext,bulk}}$, SSA and g could be calculated. The time resolution of aerosol optical parameters
was averaged to daily to match that of surface albedo.





### 3 Results and discussion

#### 3.1 Equivalent black carbon mass size distribution

##### 3.1.1 Overview

eBCMSD measured in Changzhou and Beijing was presented in Fig. 2a and Fig. 2b1 – 2b4, respectively. It could be seen that eBCMSD varied significantly and exhibited diverse patterns in both Changzhou and Beijing. For example, unimodal structure of eBCMSD occurred around December 9th 2021 in Beijing. eBCMSD did not show clear modal structure around June 2nd 2021 in Changzhou and around November 15th in Beijing. For the cases where eBCMSD exhibited modal structure, the peak diameter of the mode could change substantially with increasing pollution, such as from November 2nd 2021 to November 6th 2021 in Beijing. The peak diameter of the mode could also vary without systematical shift, such as from January 6th 2022 to January 8th 2022 in Beijing.

eBCMSD was presented with normalized probability density function (pdf) to study general characteristics of eBCMSD. Figure 5a1 and 5a2 were the normalized pdf over the whole campaign of Changzhou and Beijing, respectively. It could be seen that eBCMSD in Changzhou was significantly different from that in Beijing. There were two modes in the median of eBCMSD in Changzhou, which peaked at around 240 nm and 1249 nm, respectively. Yu et al. (2010) found 3 modes in ECMSD, namely modes around 300 nm, 1 μm and 5 μm, and named the 3 modes as condensation mode, droplet mode and coarse mode, respectively. Following the nomenclature by Yu et al. (2010), the mode peaking at 240 nm and 1249 nm could be termed as condensation mode and droplet mode, respectively. In contrast, only condensation mode was identified in median eBCMSD in Beijing, which peaked at 427 nm. The variation of eBCMSD, defined as the difference between upper quartile and lower quartile, in Changzhou was overall smaller than that in Beijing. The variation of eBCMSD in Changzhou (Beijing) ranged from 0.52 (0.54) μg m$^{-3}$ to 0.91 (1.73) μg m$^{-3}$ with average value of 0.75 (1.05) μg m$^{-3}$. The maximum upper quartile of eBCMSD in Changzhou was 1.58 μg m$^{-3}$. In comparison, the upper quartile of eBCMSD in Beijing could reached up to 2.14 μg m$^{-3}$, indicating the evolution of eBCMSD in Beijing was more drastic than that in Changzhou.

##### 3.1.2 Evolution with respect to pollution level

In order to investigate the evolution of eBCMSD under different pollution stages, eBCMSD was grouped into 3 periods: (1) clean period in which $m_{eBC,bulk}$ was lower than 0.5 μg m$^{-3}$, (2) transitional period in which $m_{eBC,bulk}$ was greater than 0.5 μg m$^{-3}$ but lower than 1.0 μg m$^{-3}$, (3) polluted period in which $m_{eBC,bulk}$ was greater than 1.0 μg m$^{-3}$. Data from clean, transitional and polluted period accounted for 22.6 % (30.9 %), 51.3 % (31.9 %) and 26.0 % (37.2 %) of total data in Changzhou (Beijing), respectively, showing that Changzhou (Beijing) was dominated by transitional (polluted) period in this study.

In the clean period, there was no distinct difference in eBCMSD between Changzhou (Fig. 5b1) and Beijing (Fig. 5b2). Neither eBCMSD in Changzhou nor eBCMSD in Beijing exhibited obvious modal structure in the size range of measurement. The value of eBCMSD in both Changzhou and Beijing decreased with increasing $D_p$ in general. For Changzhou (Beijing),



the median of eBCMSD decreased from 0.87 (0.47) μg m$^{-3}$ at 200 nm to 0.26 (0.26) μg m$^{-3}$ at 1500 nm with average value of
0.42 (0.34) μg m$^{-3}$. The variation of eBCMSD in Changzhou (Beijing) was 0.24 (0.24) μg m$^{-3}$ ~ 0.47 (0.55) μg m$^{-3}$ with
average value of 0.32 (0.35) μg m$^{-3}$, showing that the variation of eBCMSD in Changzhou was comparable to that in Beijing.

As polluted stage evolved to transitional period, the level of eBCMSD increased in both Changzhou (Fig. 5c1) and Beijing

(Fig. 5c2) compared to that in clean period. The median eBCMSD reached 0.41 (0.39) μg m$^{-3}$ ~ 1.09 (1.07) μg m$^{-3}$ with
average value of 0.75 (0.78) μg m$^{-3}$ in Changzhou (Beijing), respectively, about twice as much as the median eBCMSD in
clean period. The variation of eBCMSD in Changzhou (Beijing) reached 0.41 (0.44) μg m$^{-3}$ ~ 0.86 (0.86) μg m$^{-3}$ with average
value of 0.53 (0.61) μg m$^{-3}$, about twice as much as that in clean period. It could be seen that the value of median and variation
of eBCMSD in Changzhou were comparable to that in Beijing. However, the pattern of eBCMSD in Changzhou was obviously
different from that in Beijing. The peak value of median eBCMSD located at 240 (347) nm in Changzhou (Beijing). Median
eBCMSD in Changzhou exhibited two modes, namely condensation mode and droplet, with boundary at around 866 nm. In
comparison, median eBCMSD in Beijing only had one mode, namely condensation mode. The difference in peak diameter of
condensation mode between Changzhou and Beijing was as large as 107 nm. Median eBCMSD at clean period was subtracted
from that at transitional period to study eBC mass increment at each $D_p$, as shown in Fig. 6a1. It could be clearly seen that
mass increment in Changzhou peaked at 289 nm and 1249 nm, contributing to condensation mode and droplet mode in
eBCMSD, respectively. In contrast, mass increment in Beijing only peaked at 385 nm, contributing to condensation mode in
eBCMSD.

As the pollution stage came to polluted period, the level of eBCMSD increased drastically in both Changzhou (Fig. 5d1)

and Beijing (Fig. 5d2) compared to that in clean period. Both the level and the variation of eBCMSD increased with the
development of pollution. The median eBCMSD increased to 0.88 (0.61) ~ 2.12 (2.45) μg m$^{-3}$ with average value of 1.49
(1.52) μg m$^{-3}$ in Changzhou (Beijing), about 4 times as much as the median eBCMSD in clean period. The variation of
eBCMSD in Changzhou (Beijing) reached 0.60 (0.73) ~ 1.11 (1.06) μg m$^{-3}$ with average value of 0.92 (0.94) μg m$^{-3}$, about 3
times as much as that in clean period. The difference in pattern of eBCMSD between Changzhou and Beijing became more
distinct. Median eBCMSD in Changzhou clearly exhibited a bimodal structure where the condensation mode and droplet
mode peaked at 289 nm and 1249 nm, respectively. Median eBCMSD in Beijing exhibited a unimodal structure where the
condensation mode peaked a 527 nm. As shown in Fig. 6b1, the peak of mass increment in Changzhou (Beijing) shifted from
289 (385) nm to 347 (527) nm, varied by 58 (142) nm. The significant difference in the shift of peak indicated that aging
processes in regional background site was significantly different from that in urban site.
**3.1.3 Contribution of equivalent black carbon-containing particle larger than 700 nm to bulk equivalent black carbon**
**mass concentration**

It could be seen from Fig. 2 that eBC$_{>700}$ was ubiquitous. The median (lower quartile ~ upper quartile) of $m_{eBC,bulk}$ was 0.73

(0.52 ~ 1.03) μg m$^{-3}$ in Changzhou and 0.79 (0.43 ~ 1.31) μg m$^{-3}$ in Beijing (Fig. 7a1). The median of $m_{eBC,bulk}$ was comparable



between Changzhou and Beijing. The variation of $m_{eBC,bulk}$ in Changzhou, 0.51 µg m$^{-3}$, was smaller than that in Beijing, 0.88
µg m$^{-3}$. $m_{eBC,bulk,>700}$ in Changzhou was overall comparable to that in Beijing (Fig. 7a2). $m_{eBC,bulk,>700}$ was 0.20 (0.13 ~ 0.32)
µg m$^{-3}$ in Changzhou and 0.18 (0.10 ~ 0.33) µg m$^{-3}$ in Beijing. Considering that the variation of $m_{eBC,bulk,>700}$ in Changzhou,
0.19 µg m$^{-3}$, was comparable to that in Beijing, 0.23 µg m$^{-3}$, the larger variation in $m_{eBC,bulk}$ in Beijing was mainly from eBC-
containing particles less than 700 nm. $f_{m,>700}$ was 27.8 (20.9 ~ 36.5) % in Changzhou and 24.1 (17.5 ~ 34.2) % in Beijing (Fig.
7a3), indicating that eBC$_{>700}$ was overall one quarter of m$_{eBC,bulk}$. $f_{m,>700}$ in Changzhou was slightly larger than that in Beijing,
which was contributed by droplet mode of eBCMSD in Changzhou.

The statistics of mass contribution of eBC$_{>700}$ were studied with different pollution stages. As shown in Fig. 7a1, $m_{eBC,bulk}$

increased from 0.41 (0.33 ~ 0.45) µg m$^{-3}$ in clean period through 0.71 (0.58 ~ 0.83) µg m$^{-3}$ in transitional period to 1.33 (1.16
~ 1.71) µg m$^{-3}$ in polluted period by 3.2 times in Changzhou and increased from 0.32 (0.22 ~ 0.41) µg m$^{-3}$ in clean period
through 0.73 (0.61 ~ 0.85) µg m$^{-3}$ in transitional period to 1.47 (1.21 ~ 1.82) µg m$^{-3}$ in polluted period by 4.6 times in Beijing.
As shown in Fig. 7a2, the change of $m_{eBC,bulk,>700}$ with pollution level was substantial in both Changzhou and Beijing. For
Changzhou, $m_{eBC,bulk,>700}$ increased from 0.11 (0.07 ~ 0.15) µg m$^{-3}$ in clean period to 0.20 (0.14 ~ 0.27) µg m$^{-3}$ in transition
period, and reached 0.40 (0.29 ~ 0.50) µg m$^{-3}$ in polluted period, increasing by as large as 3.6 times from clean period to
polluted period. For Beijing, $m_{eBC,bulk,>700}$ increased from 0.07 (0.05 ~ 0.12) µg m$^{-3}$ in clean period to 0.17 (0.11 ~ 0.23) µg
m$^{-3}$ in transition period, and reached 0.36 (0.25 ~ 0.52) µg m$^{-3}$ in polluted period, increasing by as large as 5.1 times from
clean period to polluted period. The change in $m_{eBC,bulk}$ and $m_{eBC,bulk,>700}$ was overall consistent with the development of
pollution, leading to unconspicuous change in $f_{m,>700}$ (Fig. 7a3). $f_{m,>700}$ in Changzhou changed from 28.5 (20.3 ~ 36.0) % in
clean period through 28.4 (20.7 ~ 36.9) % in transitional period to 27.4 (22.6 ~ 36.2) % in polluted period. $f_{m,>700}$ in Beijing
varied from 26.2 (18.4 ~ 36.8) % in clean period through 22.8 (16.3 ~ 32.3) % in transitional period to 23.8 (18.1 ~ 31.9) %
in polluted period.
**3.1.4 Diurnal cycle**

It could be seen clearly that the level of eBCMSD during daytime was overall lower than that during nighttime in both

Changzhou (Fig. 8a1) and Beijing (Fig. 8a2), showing that eBCMSD was significantly regulated by planetary boundary layer.
For Changzhou (Beijing), eBCMSD from 10:00 to 18:00 (08:00 to 18:00) was obviously lower that from 20:00 to 06:00
(20:00 to 06:00). Accordingly, $m_{eBC,bulk}$ in Changzhou reached minimum of 0.56 (0.48 ~ 0.88) µg m$^{-3}$ at 12:00 and maximum
of 0.97 (0.80 ~ 1.24) µg m$^{-3}$ at 21:00 (Fig. 8b1). $m_{eBC,bulk}$ in Beijing reached minimum of 0.65 (0.42 ~ 1.02) µg m$^{-3}$ at 14:00
and maximum of 1.08 (0.55 ~ 1.52) µg m$^{-3}$ at 00:00, (Fig. 8b2). The apparent diurnal cycle was found in the condensation
mode of eBCMSD, which was mostly less than 700 nm. In contrast, diurnal cycle was not obvious for eBCMSD larger than
700 nm for both Changzhou and Beijing. Consequently, neither $m_{eBC,bulk,>700}$ in Changzhou (Fig. 8c1) nor $m_{eBC,bulk,>700}$ in
Beijing (Fig. 8c2) exhibited obvious diurnal cycle. $m_{eBC,bulk,>700}$ in both Changzhou and Beijing fluctuated around 0.2 µg m$^{-}$
$^{3}$, consistent with Sect. 3.1.3. Combining the diurnal variation of $m_{eBC,bulk}$ and $m_{eBC,bulk,>700}$, $f_{m,>700}$ was negatively correlated





to $m_{\text{eBC,bulk}}$ according to Eq. (3) with higher value during the daytime and lower value during the nighttime. $f_{\text{m,>700}}$ reached
maximum of 35.4 (26.6 ~ 41.1) % at 09:00 and reached minimum of 23.6 (13.9 ~ 30.8) % at 21:00 in Changzhou (Fig. 8d1).
$f_{\text{m,>700}}$ reached maximum of 31.0 (20.8 ~ 36.9) % at 15:00 and reached minimum of 23.5 (16.1 ~ 27.8) % at 01:00 in Beijing
(Fig. 8d2).
**3.2 Size-resolved absorption coefficient**
**3.2.1 Overview**
The timeseries of $\sigma_{\text{ab,size-resolved}}$ in Changzhou and Beijing were plotted in Fig. 3a and Fig. 3b1 – 3b4, respectively. $\sigma_{\text{ab,size-}}$
$_{\text{resolved}}$ varied substantially with $D_p$, time and location. In general, $\sigma_{\text{ab,size-resolved}}$ exhibited a unimodal structure with lower
value less than 5 Mm$^{-1}$ at the edge of $D_p$ spectrum and higher value larger than 20 Mm$^{-1}$ in between. The large spread of BC
absorption with respect to $D_p$ clearly highlighted the important role of particle size on absorption. The peak diameter of $\sigma_{\text{ab,size-}}$
$_{\text{resolved}}$ could vary with time. For instance, from December 9[th] 2021 to December 10[th] 2021 in Beijing and from January 22[nd]
2022 to January 25[th] 2022 in Beijing, the peak diameter of $\sigma_{\text{ab,size-resolved}}$ shifted clearly from about 400 nm to about 600 nm
and from about 500 nm to about 800 nm, respectively. The peak diameter of $\sigma_{\text{ab,size-resolved}}$ could also vary without systematical
change, such as $\sigma_{\text{ab,size-resolved}}$ in Changzhou and from January 6[th] 2022 to January 8[th] 2022 in Beijing. The complicated
variation of $\sigma_{\text{ab,size-resolved}}$ with time manifested complex mechanism influencing evolution of BC absorption.
The general characteristics of $\sigma_{\text{ab,size-resolved}}$ in Changzhou and Beijing was shown in Fig. 5a3 and Fig. 5a4, respectively.
The median $\sigma_{\text{ab,size-resolved}}$ in both Changzhou and Beijing both exhibited unimodal structure. For Changzhou (Beijing), $\sigma_{\text{ab,size-}}$
$_{\text{resolved}}$ had maximum value of 7.88 (10.59) Mm$^{-1}$ at 416.1 (427.2) nm and minimum value of 1.63 (2.90) Mm$^{-1}$ at 1500 (1500)
nm with average value of 5.39 (6.21) Mm$^{-1}$. The maximum value was 4.9 (3.7) times as large as minimum value in Changzhou
(Beijing), showing the significant dependence of absorption on particle size. $D_p$ which had higher median value of $\sigma_{\text{ab,size-}}$
$_{\text{resolved}}$ corresponded to larger variation on the whole. The variation of $\sigma_{\text{ab,size-resolved}}$ ranged from 2.25 (2.82) Mm$^{-1}$ at 1500
(1500) nm to 7.43 (17.90) Mm$^{-1}$ at 500 (527) nm with average value of 4.99 (8.97) Mm$^{-1}$ in Changzhou (Beijing). The
variation of $\sigma_{\text{ab,size-resolved}}$ was as large as the level of $\sigma_{\text{ab,size-resolved}}$ in both Beijing and Changzhou, showing the large
variability of BC absorption. The variation of $\sigma_{\text{ab,size-resolved}}$ in Beijing was overall 1.8 times as large as that in Changzhou,
indicating that the evolution of $\sigma_{\text{ab,size-resolved}}$ in different sites could be significantly different.
**3.2.2 Evolution with respect to pollution level**
$\sigma_{\text{ab,size-resolved}}$ was grouped into 3 periods based on $m_{\text{eBC,bulk}}$ as described in Sect. 3.1.2. In clean period, the value of $\sigma_{\text{ab,size-}}$
$_{\text{resolved}}$ overall decreased with increasing $D_p$ in both Changzhou (Fig. 5b3) and Beijing (Fig. 5b4), and the pattern of $\sigma_{\text{ab,size-}}$
$_{\text{resolved}}$ had no obvious modal structure. In Changzhou (Beijing), the value of $\sigma_{\text{ab,size-resolved}}$ decreased from 4.67 (3.43) Mm$^{-1}$ at
200 (427) nm to 0.88 (1.80) Mm$^{-1}$ at 1500 (1500) nm with average value of 2.95 (2.49) Mm$^{-1}$. The variation of $\sigma_{\text{ab,size-resolved}}$
in Changzhou (Beijing) ranged from 1.06 (1.57) Mm$^{-1}$ to 2.72 (3.12) Mm$^{-1}$ with average value of 2.04 (2.47) Mm$^{-1}$.
During the transitional period, the unimodal pattern could be identified in both Changzhou (Fig. 5c3) and Beijing (Fig.





5c4). Median $\sigma_{ab,size\text{-}resolved}$ peaked at 416 (427) nm with value of 7.80 (10.04) Mm$^{-1}$ in Changzhou (Beijing). Median $\sigma_{ab,size\text{-}}$
$_{resolved}$ in clean period was subtracted from that in transitional period to study absorption increment at each $D_p$, as shown in
Fig. 6a2. The increment of $\sigma_{ab,size\text{-}resolved}$ in Changzhou (Beijing) had maximum value of 3.94 (6.61) Mm$^{-1}$ at 416 (427) nm
and minimum value of 0.66 (1.15) Mm$^{-1}$ at 1500 (1500) nm. The increment of absorption was most at around 420 nm and
least at 1500 nm, showing the significant difference in the change of absorption at different $D_p$ with the development of
pollution. The maximum increment of absorption in Beijing was 1.7 times as large as that in Changzhou. Hence, the evolution
of absorption could be different substantially in different locations. The variation of $\sigma_{ab,size\text{-}resolved}$ in Changzhou (Beijing)
ranged from 1.94 (2.32) Mm$^{-1}$ to 4.03 (6.43) Mm$^{-1}$ with average value of 3.08 (4.45) Mm$^{-1}$, increasing by about 1.5 times
compared to clean period.

In the polluted period, the unimodal pattern of $\sigma_{ab,size\text{-}resolved}$ was significant in both Changzhou (Fig. 5d3) and Beijing (Fig.

5d4). Median $\sigma_{ab,size\text{-}resolved}$ peaked at 416 (527) nm with value of 16.79 (25.85) Mm$^{-1}$ and had minimum value of 2.85 (4.23)
Mm$^{-1}$ at 1500 (1500) nm in Changzhou (Beijing). Compared to transition period, peak diameter remained unchanged in
Changzhou but increased by 100 nm in Beijing, indicating the evolution of $\sigma_{ab,size\text{-}resolved}$ with aging process was different
between regional background site and typical urban site. The increment of absorption in Changzhou (Beijing) was most
significant at 416 (527) nm with value of 12.93 (22.94) Mm$^{-1}$ and least at 1500 (1500) nm with value of 1.97 (2.44) Mm$^{-1}$, as
shown in Fig. 6b2. It could be seen that the diameter of increment in absorption remain unchanged in Changzhou and shifted
by 100 nm in Beijing, indicating that absorption at different $D_p$ varied differently at different locations with the deterioration
of pollution. The variation of $\sigma_{ab,size\text{-}resolved}$ in Changzhou (Beijing) ranged from 2.19 (3.82) Mm$^{-1}$ to 9.05 (15.61) Mm$^{-1}$ with
average value of 5.72 (8.22) Mm$^{-1}$, increasing by about 3 times compared to clean period, indicating that the variability of
$\sigma_{ab,size\text{-}resolved}$ increased with the development of pollution.
**3.2.3 Contribution of equivalent black carbon-containing particle larger than 700 nm to bulk absorption coefficient**

It could be seen from the timeseries of $\sigma_{ab,size\text{-}resolved}$ in both Changzhou (Fig. 3a) and Beijing (Fig. 3b1 – 3b4) that

absorption of eBC$_{>700}$ was nonnegligible. $\sigma_{ab,bulk}$ was 4.93 (3.53 ~ 7.24) Mm$^{-1}$ in Changzhou and 6.37 (3.31 ~ 11.68) Mm$^{-1}$
in Beijing on the whole, as shown in Fig. 7b1. Both median and variation of $\sigma_{ab,bulk}$ in Changzhou were less than that in
Beijing. $\sigma_{ab,bulk,>700}$ was 1.03 (0.62 ~ 1.59) Mm$^{-1}$ in Changzhou, accounting for 19.6 (15.8 ~ 24.6) % of $\sigma_{ab,bulk}$, and 1.47 (0.81
~ 2.83) Mm$^{-1}$ in Beijing, accounting for 25.9 (19.6 ~ 33.7) % of $\sigma_{ab,bulk}$, respectively, as shown in Fig. 7b2 and Fig. 7b3. It
could be clearly seen that eBC$_{>700}$ contributed to substantial part of total absorption, and should be explicitly considered in
BC radiative estimation.

With the aggravation of pollution, the change of $m_{eBC,bulk}$ in Changzhou was overall in agreement with that in Beijing (Fig.

7a1). However, the change of $\sigma_{ab,bulk}$ with the development of pollution was different between Changzhou and Beijing (Fig.
4b1). In the clean period, $\sigma_{ab,bulk}$ in Changzhou with value of 2.71 (2.30 ~ 3.28) Mm$^{-1}$ was comparable to that in Beijing with
value of 2.47 (1.65 ~ 3.28) Mm$^{-1}$. In the transitional period, $\sigma_{ab,bulk}$ was 4.83 (4.04 ~ 6.02) Mm$^{-1}$ in Changzhou and 5.93 (4.72



~ 7.33) Mm$^{-1}$ in Beijing. The deviation in $\sigma_{ab,bulk}$ was about 1 Mm$^{-1}$ between Changzhou and Beijing. In the polluted period,
$\sigma_{ab,bulk}$ was 9.61 (7.99 ~ 11.93) Mm$^{-1}$ in Changzhou and 13.65 (10.94 ~ 17.59) Mm$^{-1}$ in Beijing. The deviation in $\sigma_{ab,bulk}$ came
to 4 Mm$^{-1}$ between Changzhou and Beijing. It could be seen that with the development of pollution, the change of $\sigma_{ab,bulk}$ in
Changzhou was less than that in Beijing. MAC$_{bulk}$, defined as the ratio of median $\sigma_{ab,bulk}$ to median $m_{eBC,bulk}$, changed from
6.61 (7.72) m$^2$ g$^{-1}$ through 6.80 (8.13) m$^2$ g$^{-1}$ to 7.23 (9.29) m$^2$ g$^{-1}$ in Changzhou (Beijing). The increase in MAC$_{bulk}$ in both
Changzhou and Beijing with the aggravation of pollution indicated the aging of BC. MAC$_{bulk}$ in Changzhou was overall lower
than that in Beijing and increased slower than that in Beijing with the development of pollution, indicating that the BC
properties and aging process in Changzhou differentiate from that in Beijing.
$\sigma_{ab,bulk,>700}$ in both Changzhou and Beijing increased with the development of pollution, as shown in Fig. 7b2. $\sigma_{ab,bulk,>700}$
increased from 0.54 (0.62 ~ 1.59) Mm$^{-1}$ through 0.96 (0.72 ~ 1.32) Mm$^{-1}$ to 1.75 (1.53 ~ 2.36) Mm$^{-1}$ in Changzhou and
increased from 0.63 (0.43 ~ 0.91) Mm$^{-1}$ through 1.36 (1.01 ~ 1.79) Mm$^{-1}$ to 3.45 (2.46 ~ 5.34) Mm$^{-1}$ in Beijing. $\sigma_{ab,bulk,>700}$
increased by 3.2 (5.5) times in Changzhou (Beijing). The relative increase of $\sigma_{ab,bulk,>700}$ was overall consistent with that of
$\sigma_{ab,bulk}$ in both Changzhou and Beijing. As a result, there was no significant change in $f_{ab,>700}$ with the development of pollution
(Fig. 7b3). $f_{ab,>700}$ varied from 19.8 (15.2 ~ 23.8) % through 19.3 (15.9 ~ 25.3) % to 19.6 (15.5 ~ 24.5) % in Changzhou and
varied from 27.9 (20.7 ~ 36.4) % through 23.2 (17.8 ~ 30.7) % to 26.7 (20.4 ~ 34.7) % in Changzhou. It could be seen that
the increase of $\sigma_{ab,bulk,>700}$ in Changzhou was less than that in Beijing with the development of pollution. Specifically,
$\sigma_{ab,bulk,>700}$ in Beijing was 2.0 times larger than that in Changzhou, showing that the change of $\sigma_{ab,bulk,>700}$ with the aggravation
of pollution could be different significantly in different sites.
**3.2.4 Diurnal cycle**
$\sigma_{ab,size\text{-}resolved}$ exhibited clear diurnal cycle in both Changzhou (Fig. 8a3) and Beijing (Fig. 8a4) with lower value of $\sigma_{ab,size\text{-}}$
$_{resolved}$ during daytime and higher value during nighttime. Accordingly, $\sigma_{ab,bulk}$ had minimum value of 3.51 (3.16 ~ 4.26) Mm$^{-}$
$^1$ at 14:00 and maximum value of 7.20 (3.80 ~ 10.58) Mm$^{-1}$ at 01:00 in Changzhou (Fig. 8b3). $\sigma_{ab,bulk}$ had minimum value of
3.96 (2.97 ~ 9.10) Mm$^{-1}$ at 14:00 and maximum value of 7.86 (4.04 ~ 13.19) Mm$^{-1}$ at 00:00 in Beijing (Fig. 8b4), reflecting
the regulation by planetary boundary layer. In contrast, neither $\sigma_{ab,bulk,>700}$ in Changzhou (Fig. 8c3) nor $\sigma_{ab,bulk,>700}$ in Beijing
(Fig. 8c4) exhibited obvious diurnal cycle. Therefore, $f_{ab,>700}$, inversely proportional to $\sigma_{ab,bulk}$, had higher value during
daytime and lower value during nighttime. For Changzhou, $f_{ab,>700}$ reached maximum at 09:00 with value of 25.3 (20.4 ~
27.4) % and came to minimum at 21:00 with value of 16.6 (13.0 ~ 19.6) % (Fig. 8d3). For Beijing, $f_{ab,>700}$ reached maximum
at 10:00 with value of 30.4 (21.1 ~ 36.3) % and came to minimum at 01:00 with value of 24.5 (17.2 ~ 28.1) % (Fig. 8d4).
**3.3 Direct radiative forcing of equivalent black carbon**
**3.3.1 Overview**
The timeseries of DRF$_{eBC}$ in Changzhou and Beijing was shown in Fig. 4a1 and Fig. 4b1 – 4b4, respectively. It could be
seen that DRF$_{eBC}$ varied significantly in both Changzhou and Beijing. DRF$_{eBC}$ was estimated to be 0.93 (0.70 ~ 1.39) W m$^{-2}$





in Changzhou and 1.10 (0.65 ~ 2.00) W m$^{-2}$ in Beijing, respectively (Fig. 7c1). The variation of DRF$_{eBC}$ was as large as the
median value of DRF$_{eBC}$, clearly indicating the large variability of BC radiative effect. DRF$_{eBC}$ increased substantially with
the aggravation of pollution (Fig. 7c1). DRF$_{eBC}$ increased from 0.38 (0.38 ~ 0.38) W m$^{-2}$ through 0.77 (0.70 ~ 0.98) W m$^{-2}$ to
1.67 (1.29 ~ 2.07) W m$^{-2}$ by 4.4 times in Changzhou and from 0.42 (0.33 ~ 0.66) W m$^{-2}$ through 1.17 (0.79 ~ 1.45) W m$^{-2}$ to
2.41 (1.68 ~ 2.86) W m$^{-2}$ by 5.7 times in Beijing with the development of pollution.
**3.3.2 Contribution of equivalent black carbon-containing particle larger than 700 nm to direct radiative forcing of**
**equivalent black carbon**
DRF$_{eBC,>700}$ was estimated to be 0.19 (0.13 ~ 0.26) W m$^{-2}$ in Changzhou and 0.20 (0.13 ~ 0.37) W m$^{-2}$ in Beijing (Fig. 7c2),
respectively, which accounted for 20.5 (18.4 ~ 22.2) % and 21.0 (16.3 ~ 26.1) % of DRF$_{eBC}$ (Fig. 7c3), respectively. Therefore,
eBC$_{>700}$ contributed to an important portion of BC radiative effect. With the aggravation of pollution, DRF$_{eBC,>700}$ increased
substantially and was different regionally (Fig. 7c2), DRF$_{eBC,>700}$ increased from 0.10 (0.10 ~ 0.10) W m$^{-2}$ through 0.17 (0.12
~ 0.26) W m$^{-2}$ to 0.24 (0.22 ~ 0.30) W m$^{-2}$ by 2.4 times in Changzhou and from 0.10 (0.08 ~ 0.12) W m$^{-2}$ through 0.20 (0.17
~ 0.24) W m$^{-2}$ to 0.47 (0.34 ~ 0.71) W m$^{-2}$ by 4.7 times in Beijing. The characteristics of $f_{DRF,>700}$ with increasing pollution
was complicated (Fig. 7c3). $f_{DRF,>700}$ varied from 25.0 (25.0 ~ 25.0) % through 21.1 (20.3 ~ 22.3) % to 17.6 (15.5 ~ 18.9) %
in Changzhou, exhibiting a decreasing trend. However, $f_{DRF,>700}$ varied from 24.4 (17.4 ~ 27.7) % through 18.4 (15.4 ~ 24.5) %
to 21.5 (19.1 ~ 26.9) % in Changzhou, without systematical change.
**3.4 Case study**
Figure 8 exhibited a pollution episode from October 31$^{st}$, 2021 to November 6, 2021 in Beijing, which was used for case
study to illustrate the large variability of eBC$_{>700}$. The mean diameter ($\overline{D}_p$) of eBCMSD was defined as
$$\log \overline{D}_p = \frac{\int \log D_p \frac{dm_{eBC}}{d\log D_p} d\log D_p}{\int \frac{dm_{eBC}}{d\log D_p} d\log D_p},$$  (12)
which was used to depict the spectral variation of eBCMSD because eBCMSD did not always had an explicit modal pattern
as mentioned in Sect. 3.1.1, and the corresponding peak diameter was not always easy to be distinguished.
With the development of pollution, $\overline{D}_p$ shifted apparently from around 400 nm to around 600 nm (Fig. 9a). $m_{eBC,bulk}$
($m_{eBC,bulk,>700}$) increased from less than 0.5 (0.15) μg m$^{-3}$ to as large as 2.5 (1.0) μg m$^{-3}$ by 5.0 (6.6) times. $\sigma_{ab,bulk}$ ($\sigma_{ab,bulk,>700}$)
increased from less than 4 (1) Mm$^{-1}$ to as large as 25 (10) Mm$^{-1}$ by 6.3 (10.0) times. DRF$_{eBC}$ (DRF$_{eBC,>700}$) increased from 1
(0.2) W m$^{-2}$ to as large as 4 (1) W m$^{-2}$ by 4.0 (5.0) times. It could be seen that the variability of eBC$_{>700}$ was significant. $f_{m,>700}$,
$f_{ab,>700}$ and $f_{DRF,>700}$ increased from about 20 %, 20 % and 20 % to as large as 50 %, 50 % and 40 %, respectively (Fig. 9b),
clearly showing important role of eBC$_{>700}$ in BC mass, absorption as well as radiative effect.
**4 Conclusions**
Black carbon (BC) mass size distribution (BCMSD) was an important factor influencing environmental and radiative effect
of BC. However, current BCMSD measurements mainly focused on BC-containing particle less than 700 nm. The





characteristics of BC-containing particle greater than 700 nm (BC$_{>700}$) remained uncertain due to limit in technique. In this
study, the characteristics of equivalent BC$_{>700}$ (eBC$_{>700}$) were measured and studied based on field measurements in eastern
China.

Equivalent BCMSD (eBCMSD) was measured from 150 nm up to 1.5 μm with time resolution of 1 hour based on the

method proposed by Zhao et al. (2022), where eBCMSD was determined by an aerodynamic aerosol classifier (AAC) in
tandem with an aethalometer (model AE33, AAC – AE33) and size-resolved particle number concentration was measured
concurrently to model the influence of particle size on mass absorption cross section (Zhao et al., 2021). AAC – AE33 was
applied to two field measurements in eastern China, namely Changzhou located in the Yangtze River Delta from May 17$^{th}$ to
June 3$^{rd}$ in 2021 and Beijing located in the North China Plain from October 29$^{th}$ 2021 to January 26$^{th}$ 2022. Changzhou was
a regional background site and Beijing was a typical urban site. The direct radiative forcing of eBC (DRF$_{eBC}$) was estimated
by Santa Barbara DISORT (discrete ordinates radiative transfer) atmospheric radiative transfer (SBDART) model (Ricchiazzi
et al., 1998).

eBCMSD was different between Changzhou and Beijing. Campaign-averaged eBCMSD in Changzhou exhibited two

modes, peaking at 240 nm and 1249 nm, respectively. In contrast, campaign-averaged eBCMSD in Beijing exhibited one
mode, peaking at 427 nm. eBC$_{>700}$ was ubiquitous in both Changzhou and Beijing. The campaign-averaged mass, absorption
as well as radiative contribution of eBC$_{>700}$ to buk eBC mass concentration ($m_{eBC,bulk}$), bulk absorption coefficient ($\sigma_{ab,bulk}$),
as well as DRF$_{eBC}$ in Changzhou and Beijing were 27.8 (20.9 ~ 36.5) % and 24.1 (17.5 ~ 34.2) %,19.6 (15.8 ~ 24.6) % and
25.9 (19.6 ~ 33.7) %, as well as 20.5 (18.4 ~ 22.2) % and 21.0 (16.3 ~ 26.1) %, respectively, manifesting the important role
of eBC$_{>700}$ in environment and climate. Both eBCMSD and size-resolved absorption coefficient ($\sigma_{ab,size\text{-}resolved}$) exhibited
diurnal variation with lower value during the daytime and higher value during the nighttime in both Changzhou and Beijing.

With the aggravation of pollution, the evolution of eBCMSD and $\sigma_{ab,size\text{-}resolved}$ in Changzhou was significantly different

from that in Beijing. The peak diameter of eBCMSD shifted from 240 (347) nm to 289 (527) nm in Changzhou (Beijing) and
the peak diameter of $\sigma_{ab,size\text{-}resolved}$ shifted from 416 (427) nm to 416 (527) nm in Changzhou (Beijing), indicating the aging
process in regional background site was distinct from that in urban site. Both the level of eBCMSD and $\sigma_{ab,size\text{-}resolved}$ increased
with the development of pollution in both Changzhou and Beijing. Accordingly, $m_{eBC,bulk}$, $\sigma_{ab,bulk}$ and DRF$_{eBC}$ in Changzhou
(Beijing) increased by 3.2 (4.6) times, 3.5 (5.5) times and 4.4 (5.7) times, respectively. $m_{eBC,bulk}$, $\sigma_{ab,bulk}$ and DRF$_{eBC}$ of eBC$_{>700}$
in Changzhou (Beijing) increased by 3.6 (5.1) times, 3.2 (5.5) times and 2.4 (4.7) times, respectively, clearly showing the
large variation of eBC$_{>700}$. Case study exhibited that contribution of eBC$_{>700}$ to $m_{eBC,bulk}$, $\sigma_{ab,bulk}$ and DRF$_{eBC}$ could increase
from 20 % to 50 %, from about 20 % to 50 % and from 20 % to 40 %, respectively. Therefore, it was highly recommended to
take BC$_{>700}$ into account in both BC field measurement and model evaluation of BC climate effect.
**Code and data availability**
The code and measurement data involved in this study are available upon request to the authors. The data involved in this study is





also available online at: https://pan.baidu.com/s/1IE2lyPg0vb8O_GPTl-dSog?pwd=pzi8.
**Author contribution**
CZ determined the main goal of this study. WZ carried experiments out and prepared the paper with contributions from all co-
authors.
**Competing interests**
The authors declare that they have no conflict of interest.

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

**Figure 1: Instrumental setup used in this study. Instruments used to measure $N_{size-resolved}$ was colored with red (green) for**
**Changzhou (Beijing).**

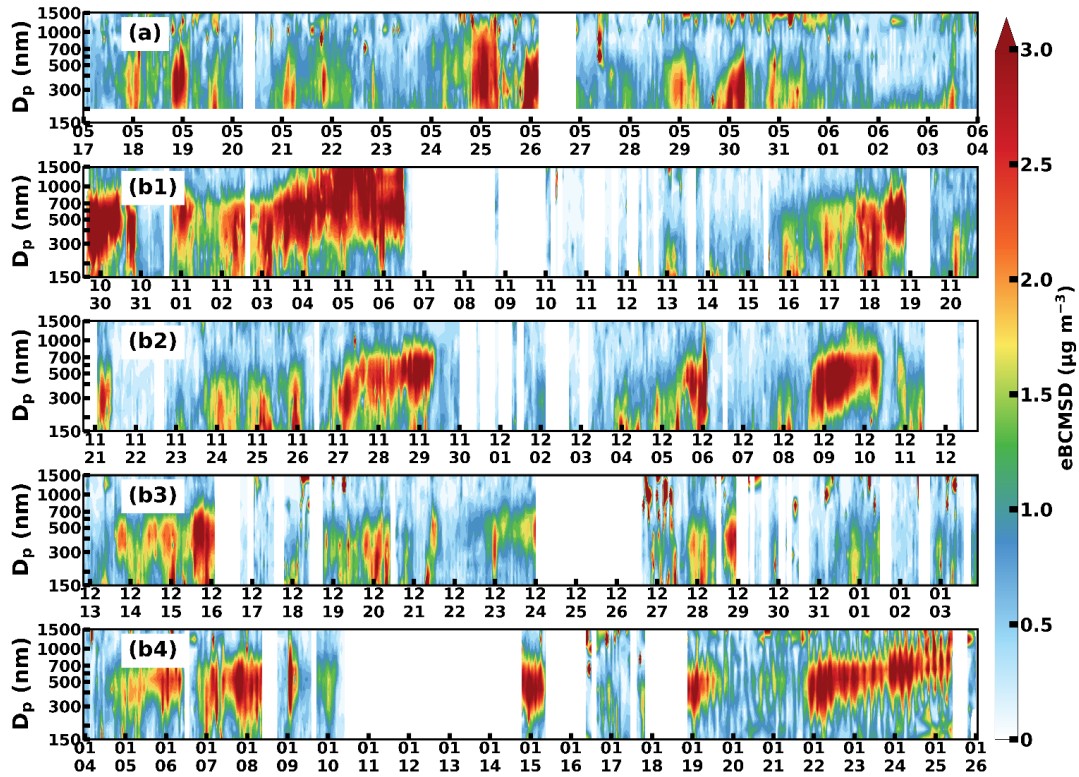


**Figure 2: Time series of eBCMSD measured in (a) Changzhou from May 17ᵗʰ 2021 to June 3ʳᵈ 2021 and (b1 – b4)**


**Beijing October 29ᵗʰ 2021 to January 25ᵗʰ 2022. (b1) to (b4) corresponded to different time ranges.**






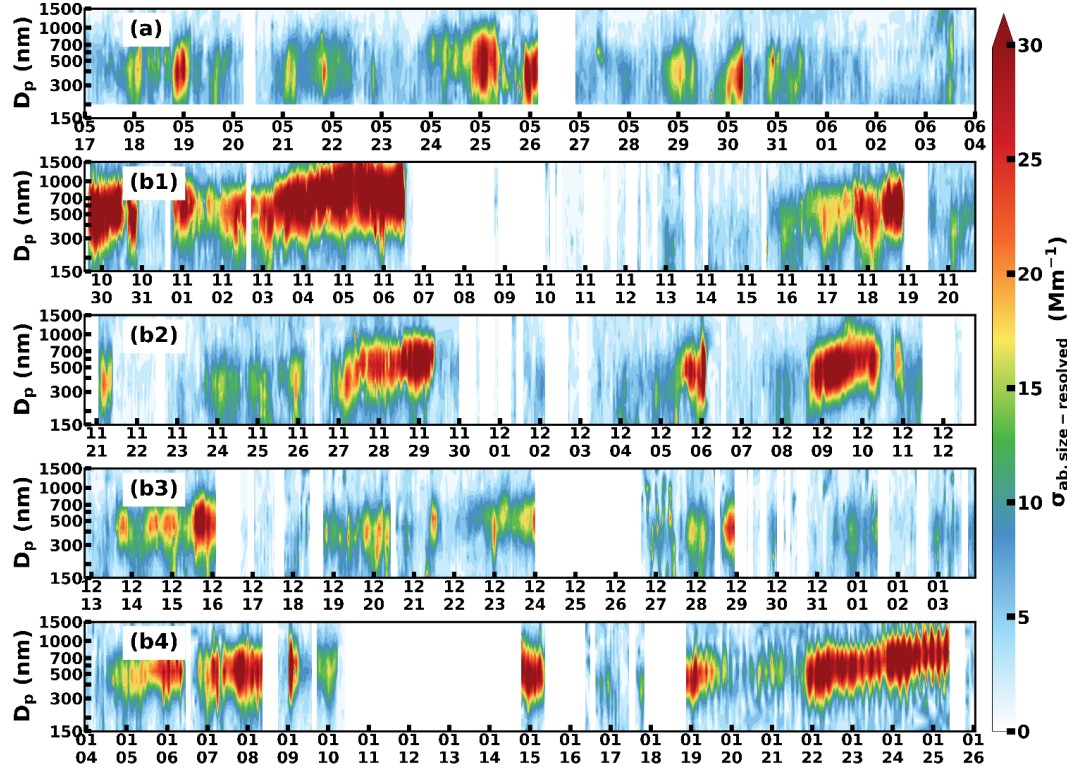


**Figure 3:** Same as Fig. 2, except for $\sigma_{ab,size-resolved}$.





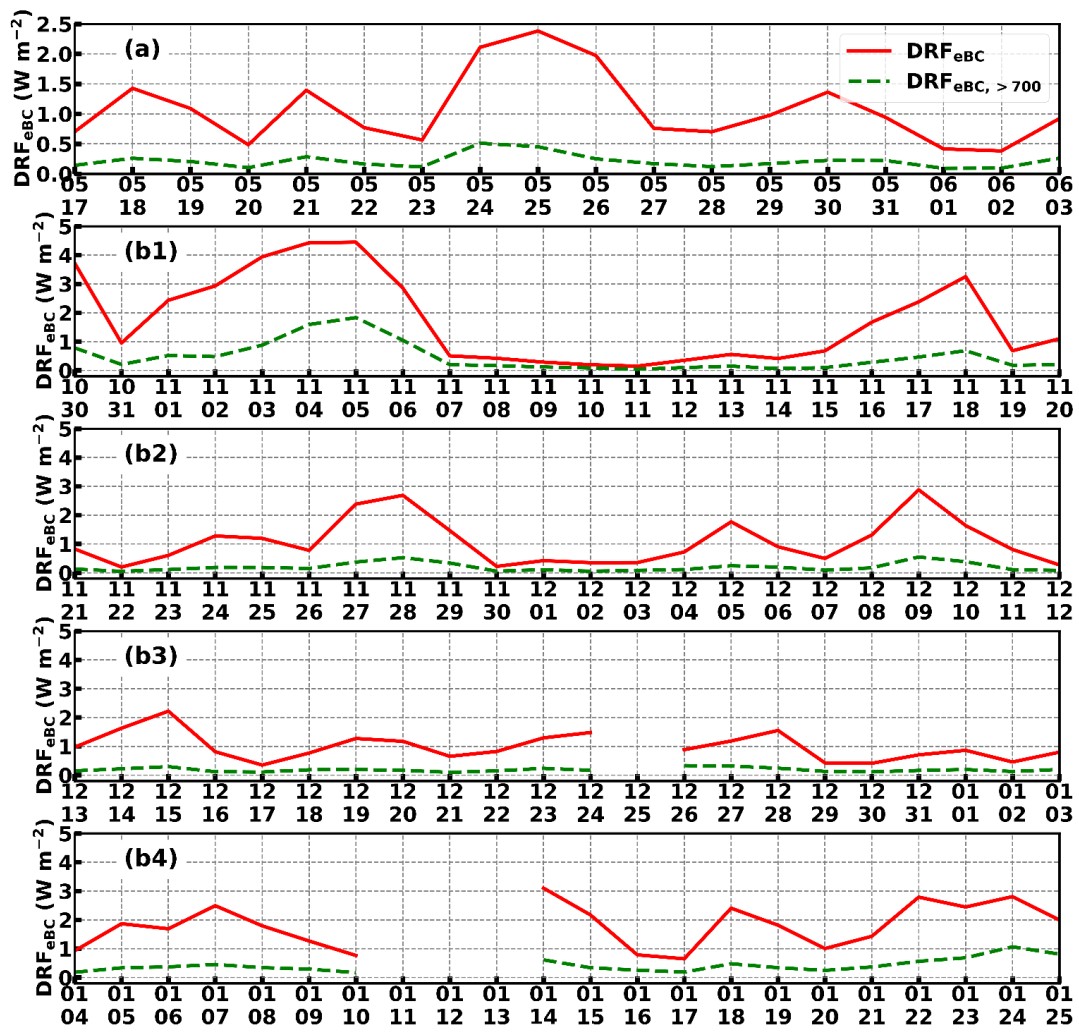


**Figure 4: Same as Fig. 2, except for DRF$_{eBC}$ (red solid line) and DRF$_{eBC,>700}$ (green dashed line).**


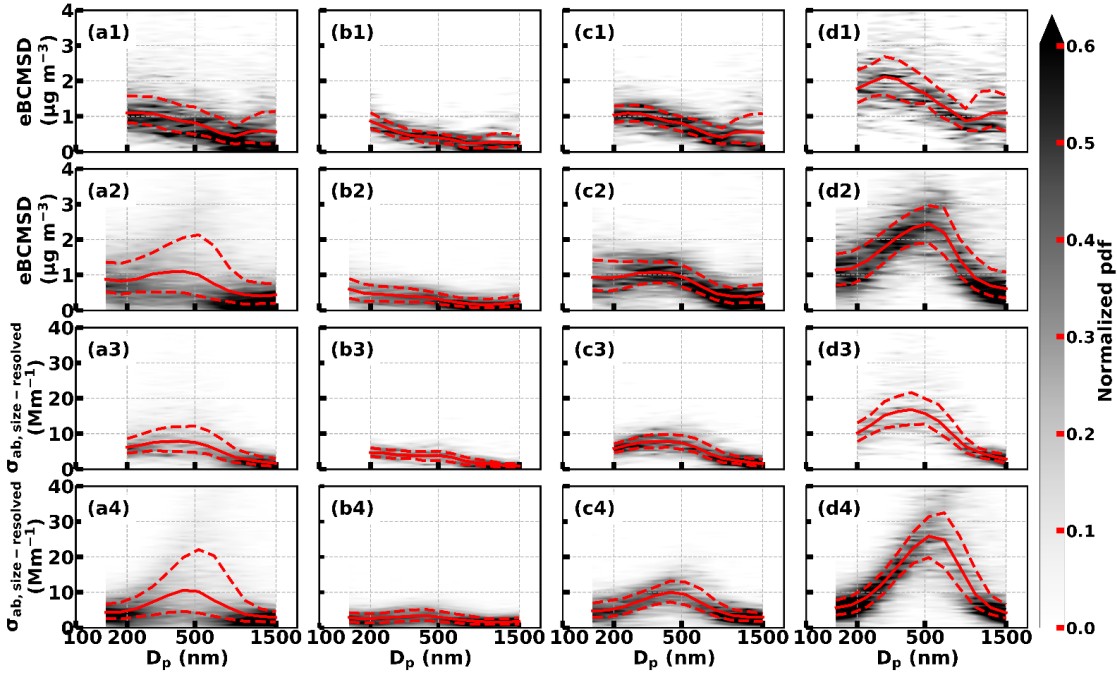

**Figure 5: Normalized pdf of eBCMSD measured in (a1 – d1) Changzhou and (a2 – d2) Beijing as well as $\sigma_{ab,size\text{-}resolved}$**

**measured in (a3 – d3) Changzhou and (a4 – d4) Beijing. (a1 – a4), (b1 – b4), (c1 – c4) and (d1 – d4) were statistics over**

**the whole campaign, clean period, transitional period and polluted period. Red solid line and red dashed lines were**

**median and lower as well as upper quartiles.**

538

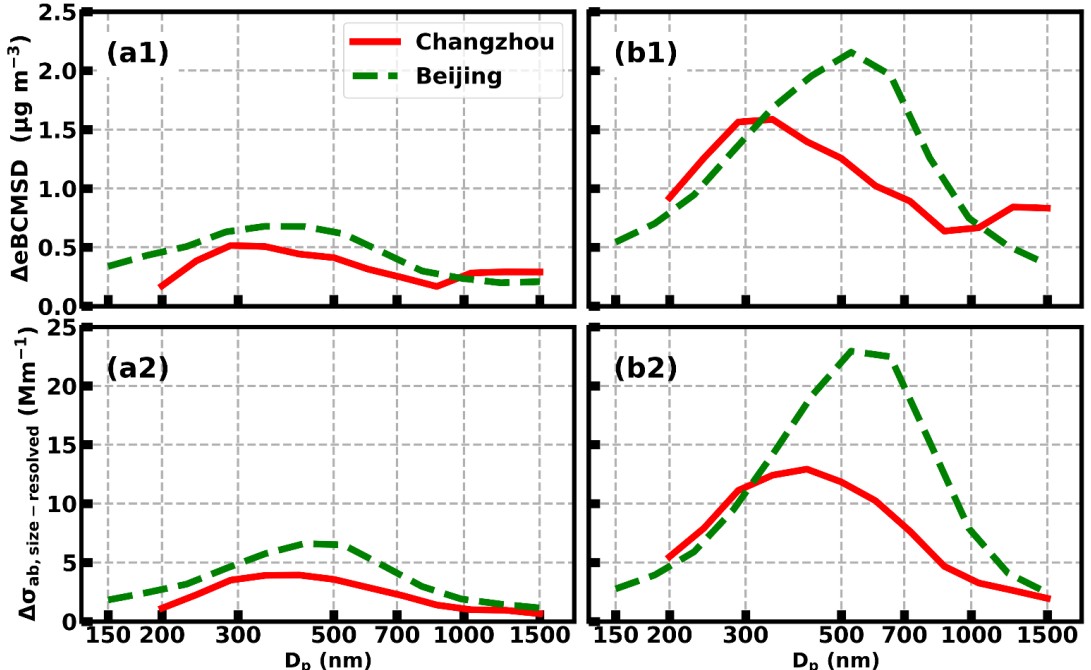

**Figure 6: Increase of median eBCMSD in (a1) transitional and (b1) polluted period relative to clean period as well as increase of median $\sigma_{ab,\text{size-resolved}}$ in (a2) transitional and (b2) polluted period relative to clean period. Red solid (green dashed) line stood for Changzhou (Beijing).**

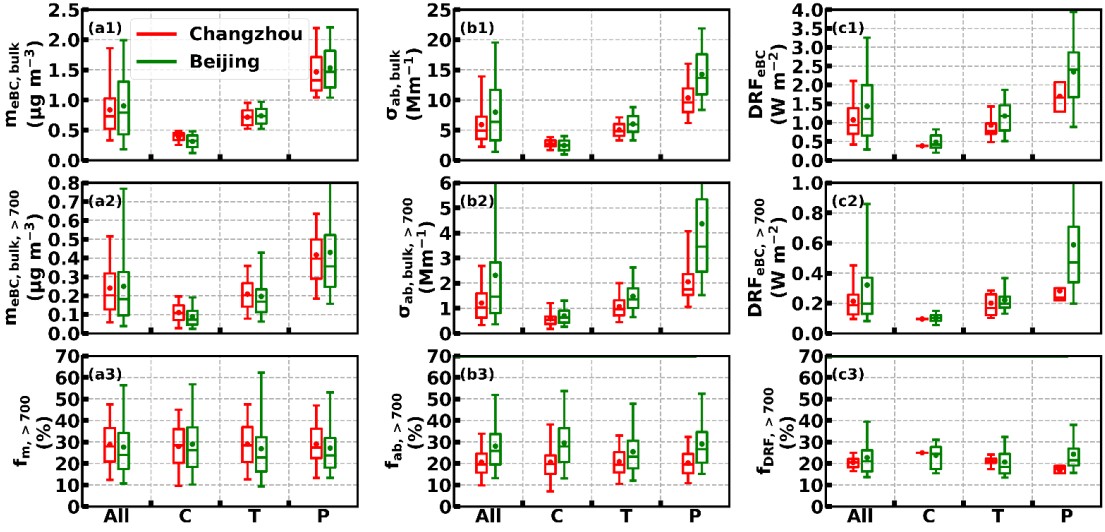

**Figure 7: Box plots of (a1) $m_{eBC,bulk}$, (a2) $m_{eBC,bulk,>700}$, (a3) $f_{m,>700}$, (b1) $\sigma_{ab,bulk}$, (b2) $\sigma_{ab,bulk,>700}$, (b3) $f_{ab,>700}$, (c1) DRF$_{eBC}$, (c2) DRF$_{eBC,>700}$ and (c3) $f_{DRF,>700}$ over the whole campaign (All), clean (C), transitional (T) as well as polluted (P) period, respectively. The box extended from the first quartile to the third quartile with a line at the median. The whiskers**





**marked 5 % and 95 % percentile. The circle inside the box was the mean value. Statistics from Changzhou (Beijing)**
**were colored red (green). The 95 percentile of $m_{eBC,bulk,>700}$ under polluted period for Beijing (a2) was 1.00 μg m$^{-3}$. The**
**95 percentile of $\sigma_{ab,bulk,>700}$ and that under polluted period for Beijing (b2) was 7.80 and 10.30 Mm$^{-1}$, respectively. The**
**95 percentile of DRF$_{eBC,>700}$ under polluted period for Beijing (c2) was 1.41 W m$^{-2}$.**

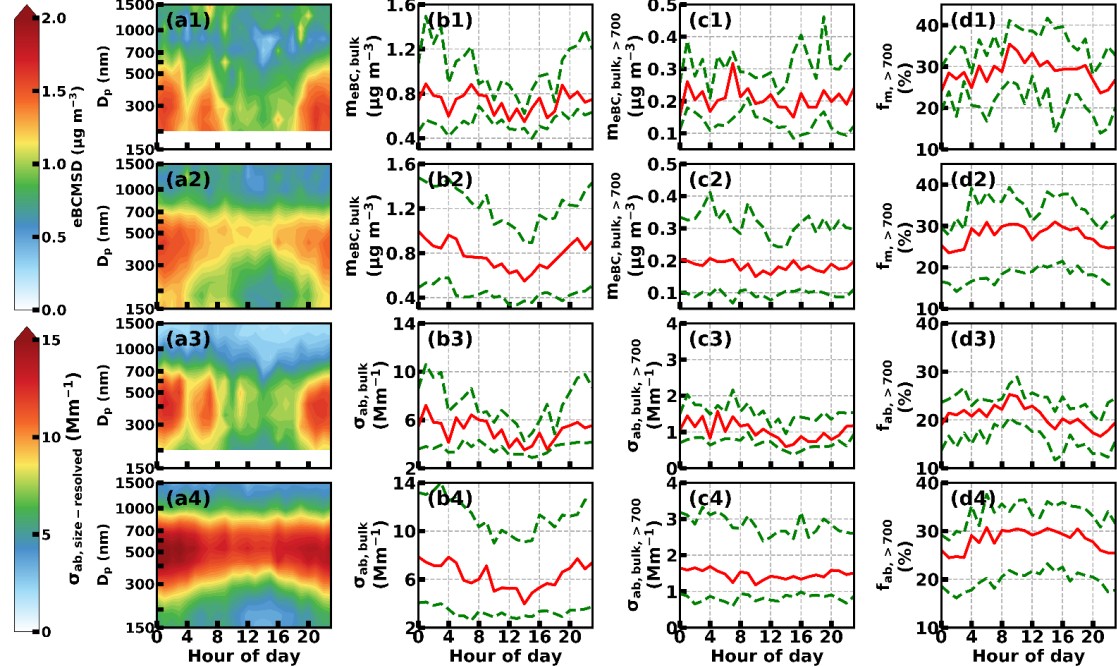


**Figure 8: Diurnal variation of (a1) eBCMSD, (b1) $m_{eBC,bulk}$, (c1) $m_{eBC,bulk,>700}$, (d1) $f_{m,>700}$ in Changzhou; (a2) eBCMSD,**
**(b2) $m_{eBC,bulk}$, (c2) $m_{eBC,bulk,>700}$, (d2) $f_{m,>700}$ in Beijing; (a3) $\sigma_{ab,size-resolved}$, (b3) $\sigma_{ab,bulk}$, (c3) $\sigma_{ab,bulk,>700}$, (d3) $f_{ab,>700}$ in**
**Changzhou and (a4) $\sigma_{ab,size-resolved}$, (b4) $\sigma_{ab,bulk}$, (c4) $\sigma_{ab,bulk,>700}$, (d4) $f_{ab,>700}$ in Beijing. Red solid line and green dashed**
**lines were median and lower as well as upper quartiles.**



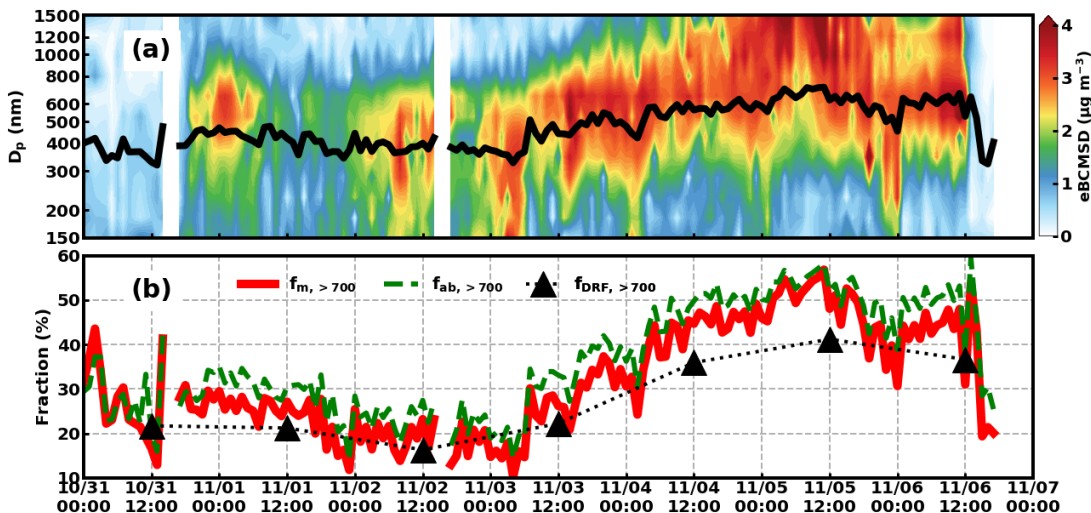

**Figure 9: (a) eBCMSD from October 31$^{st}$ 2021 to November 6$^{th}$ 2021 in Beijing and (b) the corresponding $f_{m,>700}$ (red solid line), $f_{ab,>700}$ (green dashed line) as well as $f_{DRF,>700}$ (black dotted line with triangle marker). The black solid line was $\bar{D}_p$.**