# Peer review of "Measurement report: Size-resolved mass concentration of equivalent"

_Atmospheric Chemistry and Physics, 2023_

## Author Comment (AC1)

Response to Anonymous Referee #1

**Major comments:**

*1) Throughout the paper, it is unclear whether the authors are discussing the whole diameter of the BC-containing particles or the diameter of the BC core within the BC-containing particles. In the reviewer's opinion, "BC mass size distribution (BCMSD)" usually refers to the size distribution of BC cores. However, this study seems to refer to the size distributions of whole particles containing BC and non-BC material, which is difficult to understand unless it is clearly explained. When the authors state that "the mass size distribution of BC is bimodal," the readers may assume that the mass size distribution of BC core is bimodal. In the case of this paper, can it mean that the BC core size is not bimodal, but there are two distinct groups of BC cores containing thin and thick coatings?*

Response: Thank you for your comments. In this study, BC mass size distribution (BCMSD) refers to the size distribution of whole particles containing both BC and non-BC material. We clarify it in the revised manuscript. "The mass size distribution is bimodal" means that there are two distinct groups of BC-containing particles with respect to the size of the whole particle. We make it clear in the revised manuscript. It is possible that the BC core size is not bimodal, but there are two distinct groups BC cores containing thin and thick coatings.

*2) More explanations on assumptions of mass absorption cross section (MAC) values to derive eBC mass concentration are needed, although a previous study is referred to in the manuscript. For example, even though the whole particle size is 700 nm, the MAC value should differ depending on whether the BC core is 700 nm (bare BC) or the BC core is 200 nm (thickly coated BC). If the uncertainty in the size-resolved MAC assumed in this study is large, the associated uncertainty in the size-resolved eBC mass concentrations is also considerable. Since the size-resolved absorption data is closer to the raw data, are these data more reliable and can be more central to the discussion in this paper?*

Response: Thank you so much for your comments. We add more explanation of assumptions and associated uncertainty in deriving MAC in the revised manuscript. Although size-resolved absorption data is closer to the raw data, optical parameters related to radiative transfer, such as particle scattering, have to be calculated based BC mass, and emission inventory data usually associate with BC mass, not BC absorption. Therefore, BC mass is discussed in this study rather than BC absorption.

*3) In section 2.3, the authors explain the assumptions needed for estimates of DRF, but it is unclear whether these assumptions are consistent with the observations obtained in this study. How are the temporal changes*

*in size distribution and mixing states of BC-containing particles incorporated into the estimate?*

Response: Thank you for your comments. Theses assumptions are consistent with the observations obtained in this study. Take DRF estimated at time $t_0$ of May 25th 2021 as an example, size-resolved eBC mass concentration (eBCMSD) and size-resolved particle number concentration ($N_{\text{size-resolved}}$) measured at $t_0$ are used as boundary condition at ground level to construct parameterized vertical aerosol profile. When calculating aerosol optical parameters at each level, mixing states of BC-containing particles are assumed to be the same at each level, each time and each $D_p$. With abovementioned eBCMSD, $N_{\text{size-resolved}}$ and mixing state at each level, aerosol optical parameters can then be determined based on Mie theory. Therefore, mixing state is fixed in this study. eBCMSD and $N_{\text{size-resolved}}$ are varied with level and time based on observation. We add more explanation in the revised manuscript to make it clear.

*4) Clarifications of these points and English proofreading throughout the manuscript are required.*

Response: Thanks for your comments. We make corresponding changes in the revised manuscript.
* * *
**Specific comments:**

*1) L24: "Size distribution" cannot be expressed as high or low. Concentrations are high or low.*

Response: Thanks for your comments. "lower (higher) level of eBCMSD" is changed into "smaller (bigger) value of eBCMSD".

*2) L37: "Absorption of BC increases light extinction". The wording needs to be corrected.*

Response: Thank you for your suggestion. "Absorption of BC increases light extinction" was revised into "light absorption by BC reduces atmospheric visibility".

*3) L51: "reported that" is repeated.*

Response: Thanks for your comments. One repeated "reported that" is removed.

*4) L101: In this study, BC-containing particles smaller than 200 nm are not considered when calculating the "bulk" eBC mass concentrations. However, their contribution to total eBC mass concentrations is not negligible (as suggested by Figure 2). This can lead to an overestimate of the mass fractions of eBC>700 within the total eBC mass concentrations. Clarifications are needed.*

Response: Thanks for your comments. We discuss the uncertainty in $f_{m,>700}$ associated with limited size range

in the revised manuscript.

**5) L103: D_p0 is not defined.**

Response: Thanks for your comments. $D_{p0}$ is removed in the revised manuscript.

**6) L146-149: Are these assumptions consistent with the observations obtained in this study? It is difficult to understand how the measurement values are incorporated into the DRF estimate.**

Response: Thanks for your comments. These assumptions are consistent with the observation obtained in this study. The measurement values are used to calculate aerosol optical parameters, such as aerosol extinction coefficient, asymmetry factor and single scattering albedo. These optical parameters are then feed into radiative transfer model for DRF calculation. We add more explanation in the revised manuscript to make it clear.

**7) L169: The size-resolved absorption (Figure 5a3) is monomodal, but the eBCMSD (Figure 5a1) is bimodal, indicating a strong influence of the assumed size-dependent MAC. Some more explanation should be given for deriving the size-dependent MAC.**

Response: Thanks for your comments. We add more explanation for deriving size-dependent MAC in the revised manuscript.

**8) L174-175: These are dm/dlogD values rather than eBCMSD.**

Response: Thanks for your comments. We add "value" after "eBCMSD" in the revised manuscript.

**9) L193-194: It seems obvious since the eBC mass concentration ranges for clean and transition periods were defined as such.**

Response: Thanks for your comments. This sentence is removed in the revised manuscript.

**10) L243: Any reference for interpretation or comparison? (e.g., Liu, D.et al., Contrasting physical properties of black carbon in urban Beijing between winter and summer. Atmospheric Chemistry and Physics 19, 6749-6769, https://doi.org/10.5194/acp-19-6749-2019, 2019)**

Response: Thanks for your suggestion. The study by Liu et al. (2019) offers a valuable interpretation and is cited in this study.

---

## Author Comment (AC2)

Response to Anonymous Referee #2

**Major comments:**

*1) The method section needs to be clearer to me, making me unable to validate the results. First, the AE33 measurement needs to be clarified. How did you retrieve $\sigma_{abs}$? Did you correct for multi-scattering? How do you calculate the mass fraction of BC from bulk since you do not know the MAC of bulk? How do you get MAC of eBC? How did you separate iron dust and BC? AE33 measures transmission, which is used to calculate the $\sigma$ AE33 uses the default MAC to calculate wavelength-depended BC mass concentration. Thus, I do not understand why you need to estimate the MAC of bulk to get $\sigma_{abs}$. I also read Zhao et al. 2021, but I do not know how you could use that method since you do not know the BC core size. Moreover, it is also unclear how you calculated $m_{eBC,bulk}$, and estimated the eBC fraction since you do not have an OC-EC analyzer or other measurements to measure organic and BC mass fractions. Moreover, you have different size measurements (APS, SMPS, and AAC), measuring different aerodynamic sizes (APS: optical aerodynamic size, SMPS: electrical mobility diameter, AAC: aerodynamic). How do you combine them together? Furthermore, how do you estimate the particle densities?*

Response: Thank you very much for your comments. We add the methodology of retrieving $\sigma_{abs}$ and corresponding multi-scattering correction in the revised manuscript. Multi-scattering effect was corrected in this study. We add the method of deriving MAC based on size-resolved $\sigma_{abs}$ in the revised manuscript. Dust is not considered in this study and we add this statement in the revised manuscript. MAC depends on particle size, and we derive size-resolved $m_{eBC}$ in this study. The default particle-size-independent MAC leads to inaccuracy in retriving size-resolved $m_{eBC}$. Thus, we did not use the default MAC. $m_{eBC,bulk}$ was integrated from size-resolved $m_{eBC,bulk}$. The conversion between different type of sizes was based on the study of DeCarlo et al. (2005). The particle density (1.3 g cm$^{-3}$) is a fixed value in this study based on the study by Zhao et al. (2019).

*2) The trend in mass and $\sigma\_abs$ seem statistically the same, which makes me feel your size-resolved MAC is not different across the different sizes. Please comment on that. I want to see a plot of size-resolved MAC.*

Response: Thank you for your comments. Here is an example of size-resolved MAC. The reason why mass and $\sigma_{abs}$ seem the same is that size-resolved MAC influences the relative distribution of size-resolved mass. Since the averaged size-resolved MAC (8.00 m$^2$ g$^{-1}$ in this example) is closed to the default MAC of AE33 (7.77 m$^2$ g$^{-1}$), the difference in bulk mass calculated from size-resolved MAC and default MAC is not that large.

[Figure]

**Figure 1: An example of size-resolved MAC (red solid line) and the default MAC of AE33 (7.77 m² g⁻¹).**

*3) Results and Discussion section needs to add uncertainties when you show data. Your value is very low, which might be instrument noise or data uncertainty. I suggest having a summary table.*

Response: Thank you for your suggestion. We present the uncertainties of data in the form of "median (lower quantile ~ upper quantile)" in the manuscript. A summary table (Table 1) is added to the revised manuscript.

*4) When you discuss MAC and σ_abs, you should always put the wavelength. It is meaningless to discuss light absorption properties without mentioning the wavelength.*

Response: Thanks for your comments. We add wavelength (880 nm) to MAC and σ_abs in the revised manuscript.

*5) The paper needs to be better organized. The figure number is chaotic. They should be in the same order in the text. Captions of Fig. 3 and Fig. 4 are not acceptable. You should provide all the necessary information in the caption. Also, your subplot label should always be distinct from your plots.*

Response: Thanks for your comments. We reorganize the manuscript especially the figure number so that the figure number is in the same order of the text. We provide all the necessary information in the captions of these figures.

*6) This paper needed to provide a detailed explanation of the discrepancy between Changzhou and Pekin. Is that due to different eBC sources, seasons, geographical locations, or something else? What can we learn from your results? How can we utilize your results?*

Response: Thanks for your comments. We provide detailed explanation of the discrepancy between Changzhou and Beijing in the revised manuscript. What can be learnt from our results is that BC-containing particles larger than 700 nm is of great importance and should be measured extensively. We provide a method of measuring BC-containing particles larger than 700 nm and reference values of direct radiative forcing of

BC-containing particles larger than 700 nm.

*7) The manuscript needs to be proofread by professional proofreaders before resubmission. There are lots of grammar issues.*

Response: Thanks for your comments. We make corresponding changes in the revised manuscript.
* * *
**Specific comments:**

*1) L56-59, "It should be noted … (chow et al., 2001). I suggest discussing "Soot superaggregates from flaming wildfires and their direct radiative forcing." Offline microscopy technology can be used to analyze large soot particles.*

Response: Thanks for your suggestions. We add discussion with respect to "Soot superaggregates from flaming wildfires and their direct radiative forcing".

*2) Figure 5 a2 and a4: Why PDF mode overlaps with lower quartiles, not the median?*

Response: Thanks for your comments. There are quite a few values are larger than the lower quartiles. The distribution of these larger values is sparser and less concentrated, making them less distinguishable than the PDF mode in the plots but still influencing the position of median.

*L179-181, "In order to … 1.0 µg m-3." How did you define the thresholds of these three periods? Based on statistical analysis or literature?*

Response: Thank you for your comments. The thresholds are defined based on statistical analysis on the data. By this definition, the data in three periods are roughly 1:1:1, making the amount of data relatively unbiased for these three periods.

*L218, "It could be … ubiquitous." When you say something is ubiquitous, abundant, etc., you should show the fraction with uncertainties. Please check the manuscript and revise it accordingly.*

Response: Thank you for your suggestions. We make corresponding revision.

*L242-243, "It could be … boundary layer." What do you mean by Level of eBCMSD? Total mass or particle size? Why do you say that the planetary boundary layer regulates it? There should be more emission of eBC during the daytime, especially at morning and evening traffic peaks.*

Response: Thank you for your suggestions. The level of eBCMSD means the value of eBCMSD. We change "level" into "value" in the revised manuscript to avoid ambiguity. We consider the influence of diurnal emission of eBC in the revised manuscript.

***L260-261, "The large spread … on absorption." How about different BC mass in different size bins?***

Response: Thank you for your comments. It is possible that BC mass influence absorption in different size bins. But we removed the paragraph for better organization of the paper.

***L320-323, "The increase in … that in Beijing." Please explain the difference in detail (see my previous major comment).***

Response: Thank you for your comments. We add more description on the two measurement sites in the revised manuscript.

***Section 3.4 Case study. Why do you have this section here? It should be discussed in your previous sections. Moreover, what's the advantage of using mean diameter compared with geometric mean diameter?***

Response: Thanks for your comments. Case study is moved to previous section. The mean diameter is actually geometric mean diameter. We changed "mean diameter" into "geometric mean diameter".

***The conclusions section needs to add more discussion about environmental implications.***

Response: Thanks for your comments. We add more discussion about environmental implications.

**References**

DeCarlo, P. F., Slowik, J. G., Worsnop, D. R., Davidovits, P., and Jimenez, J. L.: Particle morphology and density characterization by combined mobility and aerodynamic diameter measurements. Part 1: Theory, Aerosol Science and Technology, 39, 184-184, 10.1080/02786820590928897, 2005.

Zhao, G., Tao, J. C., Kuang, Y., Shen, C. Y., Yu, Y. L., and Zhao, C. S.: Role of black carbon mass size distribution in the direct aerosol radiative forcing, Atmospheric Chemistry and Physics, 19, 13175-13188, 10.5194/acp-19-13175-2019, 2019.

---

## Author Response (AR2)

Response to Anonymous Referee

**Technical corrections:**

*1) L19: "The results revealed that the value of eBCMSD in both Changzhou and Beijing increased with increasing pollution." The meaning of "the value of eBCMSD" is not clear. Does it mean mass concentration level or particle size? If it means concentration, it seems obvious that it increases with increasing pollution. Please consider the revision.*

Response: Thank you for your comments. The meaning of "the value of eBCMSD" is mass concentration level. This sentence is removed in the revised manuscript.

*2) L118: "at could by". Please correct this sentence.*

Response: Thanks for your comments. This sentence is corrected in the revised manuscript.

*3) L120: "Where" should be "where".*

Response: Thanks for your comments. "Where" is changed to "where" in the revised manuscript.

*4) L188: Please specify the wavelength for these refractive indices.*

Response: Thank you for your comments. The wavelength for these refractive indices is specified in the revised manuscript.